# The Mandible Ameliorates Facial Allograft Rejection and Is Associated with the Development of Regulatory T Cells and Mixed Chimerism

**DOI:** 10.3390/ijms222011104

**Published:** 2021-10-14

**Authors:** Dante De Paz, Ana Elena Aviña, Esteban Cardona, Chin-Ming Lee, Chia-Hsien Lin, Cheng-Hung Lin, Fu-Chan Wei, Aline Yen Ling Wang

**Affiliations:** 1Department of Plastic Surgery, Chang Gung Memorial Hospital, Taoyuan 333, Taiwan; dantelio27@hotmail.com (D.D.P.); aviestlani@hotmail.com (A.E.A.); lukechlin@gmail.com (C.-H.L.); fuchanwei@gmail.com (F.-C.W.); 2Center for Vascularized Composite Allotransplantation, Chang Gung Memorial Hospital, Taoyuan 333, Taiwan; aimin0703@gmail.com (C.-M.L.); qennel@gmail.com (C.-H.L.); 3Department of Head and Neck Surgery, National Police Hospital, Lima 15072, Peru; 4Department of Plastic Surgery, Clínica IPS Universitaria León XIII, University of Antioquia, Medellín 050010, Colombia; estebancardona@icloud.com; 5College of Medicine, Chang Gung University, Taoyuan 333, Taiwan

**Keywords:** vascularized composite allotransplantation, facial transplantation, mice, mandible, regulatory T cells, mixed chimerism

## Abstract

Vascularized composite allografts contain various tissue components and possess relative antigenicity, eliciting different degrees of alloimmune responses. To investigate the strategies for achieving facial allograft tolerance, we established a mouse hemiface transplant model, including the skin, muscle, mandible, mucosa, and vessels. However, the immunomodulatory effects of the mandible on facial allografts remain unclear. To understand the effects of the mandible on facial allograft survival, we compared the diversities of different facial allograft-elicited alloimmunity between a facial osteomyocutaneous allograft (OMC), including skin, muscle, oral mucosa, and vessels, and especially the mandible, and a myocutaneous allograft (MC) including the skin, muscle, oral mucosa, and vessels, but not the mandible. The different facial allografts of a BALB/c donor were transplanted into a heterotopic neck defect on fully major histocompatibility complex-mismatched C57BL/6 mice. The allogeneic OMC (Allo-OMC) group exhibited significant prolongation of facial allograft survival compared to the allogeneic MC group, both in the presence and absence of FK506 immunosuppressive drugs. With the use of FK506 monotherapy (2 mg/kg) for 21 days, the allo-OMC group, including the mandible, showed prolongation of facial allograft survival of up to 65 days, whereas the myocutaneous allograft, without the mandible, only survived for 34 days. The Allo-OMC group also displayed decreased lymphocyte infiltration into the facial allograft. Both groups showed similar percentages of B cells, T cells, natural killer cells, macrophages, and dendritic cells in the blood, spleen, and lymph nodes. However, a decrease in pro-inflammatory T helper 1 cells and an increase in anti-inflammatory regulatory T cells were observed in the blood and lymph nodes of the Allo-OMC group. Significantly increased percentages of donor immune cells were also observed in three lymphoid organs of the Allo-OMC group, suggesting mixed chimerism induction. These results indicated that the mandible has the potential to induce anti-inflammatory effects and mixed chimerism for prolonging facial allograft survival. The immunomodulatory understanding of the mandible could contribute to reducing the use of immunosuppressive regimens in clinical face allotransplantation including the mandible.

## 1. Introduction

Most reconstructive procedures for major tissue defects such as trauma, tumor excision, or congenital anomalies can be performed with autologous tissue. The property of spontaneous regeneration and self-bone healing can be used in maxillo-facial traumatology to treat unilateral mandibular subcondylar fractures using an acrylic splint for repositioning mandibular function [1]. Both bone compaction and osseodensification techniques have been demonstrated to enhance the primary implant stability in dental implantology [2], which is critical for edentulous jaw recovery [3]. The treatment of orofacial anomalies such as orofaciodigital syndrome, which is a group of hereditary diseases characterized by deformities of the face, mouth, feet, and hands, involves careful periodic follow-up and a multidisciplinary approach [4]. Other orofacial anomalies, such as temporomandibular diseases, including multifactorial musculoskeletal diseases related to masticatory muscles, the temporomandibular joint, and accessory structures, can be treated by physiotherapy such as massage techniques [5,6] or by polyphenols like curcumin, resveratrol, and epigallocatechin3-gallate [7]. Coronoid process hypertrophy restricts mouth opening because of an abnormal volumetric increase of the mandibular coronoid process, which can be treated using physical therapy and transcutaneous electrical nerve stimulation to improve the masticatory muscle condition and decrease muscle tension, respectively [8]. However, large defects or complex injuries are not suited to conventional treatment, especially if the bone is injured or lost because of trauma or tumor removal surgery. A composite allograft including the bone is an alternative therapy for bone defects. Therefore, vascularized composite allotransplantation (VCA) shows great promise for patients requiring a hand [9,10,11,12,13] or face transplant [14,15,16,17,18,19,20,21,22,23,24,25], by aiding the restoration of normal and functional tissue grafts [26,27]. Lifelong immunosuppression is required and often results in tumor risks and opportunistic infections. If the need for lifelong immunosuppressive drugs could be obviated or decreased, then VCA could be more widely used.

VCA differs from organ transplantation, involving a specific type of cell. A composite graft contains different kinds of cells, such as skin, muscle, bone, and nerve tissues. Different tissues cause different degrees of alloimmune rejection [28]. For example, single limb-related tissue, such as skin, bone, muscle, or blood vessels, elicit a relatively lower humoral immunity, in that order [28]. Muscle, skin, bone, or vessels elicit relatively lower cellular immunity, in that order. Therefore, the skin has the most antigenicity derived from limb tissue, whereas vessels have the least antigenicity of these individual limb tissues. VCA tissues especially have higher antigenicity compared with organ allografts [29,30]. One study demonstrated that comparing skin tissue with other organ transplantations, the skin and lungs displayed the most antigenicity, the liver and heart exhibited less antigenicity, and the kidney and pancreas had the least antigenicity [29]. Facial allotransplantation elicits a more complex and aggressive rejection process [23,30,31]. Facial models have been established in primates, swine, sheep, canines, rabbits, and murine animals for the transplantation of the mystacial pad, mandibular osteomyocutaneous, mandibular hemijoint, hemiface with mandible and tong, hemiface with ear and scalp, mycutaneous hemiface, and ear, and for further study of facial transplant immunology [26]. Other composite grafts, such as hindlimb, abdominal wall, penile, and uterus, etc., have also been established in various animal transplantation models [26]. However, different kinds of facial tissues elicit different degrees of rejection. To investigate strategies for achieving facial allograft tolerance, we established a mouse facial transplant model including skin, muscle, the mandible, teeth, oral mucosa, and vessels for future facial VCA studies [32].

Magnetic resonance has shown bone marrow hematopoietic stem cells in the mandible [33,34]. Data have demonstrated that orofacial bone marrow-derived mesenchymal stem cells (BM-MSCs) are a distinctive MSC population and play an important regulatory role in systemic immunity, especially increased osteogenic potential to enhance bone formation [35,36]. These results suggest that the mandible may be able to induce mixed chimerism and anti-inflammation [37,38,39,40]. The transplantation of bone marrow hemopoietic stem cells is well known to establish mixed chimerism, where the immune cells of the donor and recipient differentiated from bone marrow hematopoietic stem cells can coexist and achieve a balance in the host [41,42,43,44,45,46]. Moreover, the induction of mixed chimerism led to allograft tolerance without the lifelong administration of immunosuppressive drugs [47,48,49]. MSC-mediated induction of anti-inflammatory immune responses also inhibits allograft rejection [40,50,51,52,53,54,55,56,57,58,59], which may be achieved by the orofacial BM-MSCs. Therefore, the mandible is hypothesized to have a critical modulation role in the process of facial allograft rejection. To further understand the effects of the mandible on facial allograft survival, we compared the facial osteomyocutaneous (OMC) group with the mandible and the myocutaneous (MC) group without the mandible in this study.

## 2. Results

### 2.1. Surgery in the Facial Osteomyocutaneous and Myocutaneous Grafts

We established the mouse hemiface transplant model, which comprises the skin, muscle, mandible, oral mucosa, and vessels and can be used for VCA studies [32]. The placement of the transplanted hemiface on the neck can reduce the risk of autonomy (mice biting an insensate flap) and increase surgery success. The cuff technique was utilized for vessel anastomosis [60,61]. To further understand the effects of the mandible on facial allograft survival, we removed the mandible from a whole hemiface subunit, including the skin, muscle, mandible, teeth, oral mucosa, and vessels, in the MC group, compared to the OMC group with the whole hemiface (Figure 1A). Representative charts of donor and recipient surgeries in the two groups are shown in Figure 1.

### 2.2. The Mandible Assists Facial Allograft Survival

To investigate the effects of the mandible on facial allograft survival, we compared the survival rate of the OMC allograft with the mandible and that of the MC allograft without the mandible in the presence or absence of an immunosuppressive regimen. A hemiface from a BALB/c mouse, including the skin, muscle, mandible, teeth, oral mucosa, and vessels, was grafted onto the neck of a C57BL/6 mouse as an allogeneic face transplantation. Then, a C57BL/6 donor face was grafted onto a C57BL/6 mouse as a syngeneic face transplantation. A 100% survival rate was observed for both the OMC and MC grafts in the syngeneic group. Without any immunosuppressive regimen, rapid aggressive necrosis leading to facial allograft rejection occurred in the MC allograft before POD 9. In the allogeneic OMC (allo-OMC) group, a similar rate of rejection and delayed onset were observed, resulting in a slightly extended survival of 12 days (Figure 2A). With a low dosage of immunosuppressive FK506 drugs, the allo-OMC group showed a similar rejection rate and delayed onset, resulting in a significantly extended survival of 42 days compared with the allogeneic MC (allo-MC) group (28 days). Interestingly, with a high dosage of immunosuppressive FK506 drugs, the allo-OMC group showed a slow rejection rate and delayed onset, resulting in a significantly extended survival of 65 days compared with the allo-MC group (34 days) (Figure 2B). This demonstrates that the mandible has a regulatory role in the prolongation of facial allograft survival, irrespective of the presence or absence of immunosuppressive regimens.

### 2.3. Facial Allograft with the Mandible Exhibits Decreased Lymphocytic Infiltration

To further evaluate the regulatory role of the mandible in facial allograft survival, the degree of lymphocyte infiltration in the allograft was evaluated on POD 14 using histological H&E staining. The allo-OMC group had less lymphocyte infiltration, whereas the allo-MC group exhibited significantly increased numbers of lymphocytes within the dermis, epidermis, and muscle, leading to in severe tissue destruction (Figure 3). This finding indicated that the facial allograft including the mandible exhibited a decreased level of lymphocyte infiltration, resulting in a prolonged allograft survival.

### 2.4. The Mandible Facilitates the Induction of Anti-Inflammatory Responses

To investigate the effects of the mandible on alloreactive lymphocyte developments in the hemiface allotransplanted mice, we analyzed CD4^+^ T, CD8^+^ T, CD19^+^ B cells, NK1.1^+^ nature killer cells (NKs), CD11c^+^ dendritic cells (DCs), and CD11b^+^ macrophages using flow cytometry. Incoming alloantigens are taken up and processed by DCs and presented to T cells for T cell stimulation. The activated CD4^+^ T cells and autoantibody-producing plasma cells from naïve B cell differentiation lead to alloimmune responses and finally result in allograft loss. NKs, which secrete INF-γ, mediate innate immune responses. The priming effects of IFN-γ induce macrophages to produce higher levels of pro-inflammatory cytokines, but lower levels of anti-inflammatory cytokines [62]. The peri-allograft lymphoid system, blood, spleen, and lymph nodes (LNs) were harvested on POD 14. Both allo-OMC and allo-MC groups showed similar percentages of T and B cells in the three lymphoid organs. The allo-OMC group showed a decreased trend of NKs, DCs, and macrophages in the blood, and a reduced trend of NKs and DCs in LNs, but displayed similar percentages of these cells in the spleen compared with the allo-MC group (Figure 4A).

To further evaluate the effects of the mandible on alloimmune responses in hemiface allotransplanted mice, we analyzed cytokine-producing CD4^+^ T cells using flow cytometry. Cytokines can be divided into two groups: pro-inflammatory and anti-inflammatory. Different T cell subsets possess different unique properties and functions [63,64]. Pro-inflammatory IFN-γ cytokine-producing T helper 1 (Th1) and anti-inflammatory cytokine-producing regulatory T cells (Tregs) were examined. The allo-OMC group exhibited a significant decrease in Th1 cells in the blood and LNs, but showed a similar percentage of Th1 cells in the spleen, compared to the allo-MC group. Interestingly, the allo-OMC group displayed a significant increase in the percentage of Tregs in LNs and a slight increase in the percentage of these cells in the blood and spleen compared with the allo-MC group (Figure 4B). This finding suggests that the mandible is associated with the formation of anti-inflammatory immune responses.

### 2.5. The Mandible Is Associated with the Induction of Mixed Chimerism

The induction of mixed chimerism was demonstrated to prolong allograft survival and even result in lifelong allograft tolerance in the host. It refers to the coexistence of donor and recipient immune cells. The mandible has been shown to possess bone marrow hematopoietic stem cells [34,35], which may differentiate into donor immune cells in the recipient. Thus, the allo-OMC group-mediated allograft survival may be correlated with the induction of mixed chimerism. To assess the levels of mixed chimerism in the hemiface-allotransplanted mice, donor immune cells were analyzed using flow cytometry. H2^d^ antibody was used to distinguish donor cells from recipient cells because H2^d^ class I histocompatibility antigen was restricted to the BALB/c donor cells. On POD 14, leucocytes were harvested from the blood, spleen, and LNs of both allogeneic groups. Significantly increased percentages of donor immune cells were seen in the blood, spleen, and LNs of the allo-OMC group compared with the allo-MC group (Figure 5A).

To further investigate whether donor hematopoietic stem cells differentiate into various lymphocyte and monocyte subsets, CD4^+^ T, CD8^+^ T, and CD19^+^ B cells, NK1.1^+^ NKs, CD11c^+^ DCs, and CD11b^+^ macrophages of donor cells were evaluated using flow cytometry. Donor immune cells in the allo-OMC group displayed the priority to differentiate into B cells, followed by DCs and macrophages, and then T cells and NKs (Figure 5B). This finding suggests that the mandible may facilitate the induction of mixed chimerism and further multi-lineage chimerism developments.

## 3. Discussion

Compared with the femur, the mandible has a different embryonic origin and thus distinctive osteogenic potential from bone marrow stromal cells [35,36,65], distinctive recovery from irradiation [66], distinctive differentiation rates from hematopoietic stem cells [67], and distinctive expression of endothelial growth factor [68]. The mandible development comes from the neural crest cells of the neuroectoderm germ layer and undergoes intramembranous ossification [68]. Although BM-MSCs isolated from the mandible and femur both differentiate into osteoblasts, mandible-derived BM-MSCs significantly enhance osteoblast gene expression, mineralization, and alkaline phosphatase activity. Compared with femur BM-MSCs, implanted BM-MSCs from the mandible differentiate with 3-fold greater mineralization and 70% larger bone nodules [35,36,65]. Mandible BM-MSCs have demonstrated a higher survival and recovery compared with iliac crest BM-MSCs through proliferation, clonogenic survival, and cell-cycle assays [66]. However, the critical mediator of angiogenesis and osteogenesis, such as vascular endothelial growth factor, showed reduced expression levels for mRNA and proteins in the mandible compared with the femur [68].

In this study, we found that the allo-OMC group (facial allograft including the mandible) showed prolonged survival of the allograft, decreased lymphocytic infiltration, and decreased pro-inflammatory Th1 cells, but increased anti-inflammatory Treg cells in LNs (Figure 4). This finding suggests that the mandible has the immunomodulatory ability to regulate alloimmune responses and the balance between anti-inflammatory and pro-inflammatory processes for allograft survival. Reports are increasingly demonstrating that MSCs have immunosuppressive properties through tolerogenic mediator secretion, such as hepatocyte growth factor, interleukin-10, and transforming growth factor-β, to inhibit lymphocyte proliferation [37,39,40]. Anti-inflammatory interleukin 10 and transforming growth factor-β cytokines can induce the development of Tregs and inhibit the proliferation of effector T cells, such as Th1 cells. Therefore, mandible-induced anti-inflammatory immune responses may be associated with the immunosuppressive function of MSCs. Moreover, the immunomodulatory property of MSCs is also correlated with an increased level of nitric oxide production, mediated by IFN-γ through enhanced inducible nitric oxide synthase expression in MSCs [69]. Evidence shows that mouse MSCs derived from the mandible have a strong suppressive effect on anti-CD3 antibody-induced T-cell proliferation, along with increased production of nitric oxide, when activated with IFN-γ [35]. Therefore, MSCs in the mandible may facilitate anti-inflammatory responses and subsequently, facial allograft survival.

In addition, significant differences were observed for decreased Th1 and increased Treg cells in LNs of the allo-OMC group, but not in the spleen and blood. This may be possible for LNs near the facial allograft, which are most reactive to alloantigens and are the closest to the alloantigens of the allograft when revascularization of the allograft occurs. Thus, the LNs contribute to the mounting of alloimmunity targeted against alloantigens, triggering the rejection process and then allograft destruction. Allogeneic mandibular MSCs may modulate the lymphocyte populations and balance between anti-inflammatory and pro-inflammatory processes in LNs near the facial allograft.

The induction of mixed chimerism can result in allograft tolerance without lifelong administration of immunosuppressive regimens [70,71]. Mixed chimerism, defined as a state that is composed of host and donor immune cells, and achieves a balance through both allogeneic and autologous hematopoietic differentiation systems. This state is usually attained through bone marrow cell transplantation. It is well established that mixed chimerism that occurs after bone marrow transplantation results in transplantation tolerance for other organs or tissue from the same donor. We found that allogeneic mandibular bone marrow has the ability to induce mixed chimerism and multilineage chimerism developments (Figure 5), which is similar to the findings reported for rat hemiface/tongue/mandible allografts associated with chimerism induction [72]. Allogeneic mandibular bone marrow may regulate the mixed chimerism state and subsequently be responsible for facial allograft survival.

Mice are suitable for the research of transplant immunology due to various immunological antibodies, immunodeficient mice, or transgenic mice for use. The small diameter of the mouse vessels, around 0.2–0.4 mm, makes vascular anastomosis in face transplantation more difficult if we use hand-suturing. To overcome the issue, we used the cuff technique to perform the vascular anastomosis in mice between the donor and the recipient. Moreover, a mouse only has 2 c.c. blood volume. Once the mouse loses massive blood during the surgery operation, it will easily cause a failed operation and even mouse death. Thus, we performed a small number of mouse surgery for the survival observation of each allograft group in the study. However, face transplantation is a unique reconstruction strategy, which results in satisfactory functional recovery and aesthetics. Motor and sensory nerves can also be functionally restored. Unfortunately, face transplantation still faces many challenges such as lifelong immunosuppression and uncertain long-term results. It may have more serious problems than hand transplants such as possible loss of vision, chronic midface infections, feeding diseases, or chronic tracheal complications. If these challenges can be overcome, face transplantation definitely assists more patients and improves their self-confidence and quality of life.

## 4. Materials and Methods

### 4.1. Mice

BALB/c and C57BL/6 mice (8–10 weeks old) were purchased from the National Laboratory Animal Center, Taiwan and were used as donors and recipients, respectively. Postoperative animal care included analgesia with ketoprofen (2.5 mg/kg/d) and antibiotic prophylaxis with cefazolin (50 mg/d) subcutaneously for 3 days. The mice were kept in an enriched environment with an abundance of nesting materials. The mouse procedure performed in the study was in full compliance with the recommendations specified in the Chang Gung Memorial Hospital′s Animal Research Guidelines for the Care and Use of Laboratory Animals. The mouse protocol was approved by the Committee on the Ethics of Animal Experiments of the Chang Gung Memorial Hospital (CGMH) in Taiwan and Institutional Animal Care and Use Committees (IACUC) of CGMH in Taiwan under permit numbers IACUC 2018080701(approved on 5 September 2018) and IACUC 2019120201 (approved on 1 April 2020).

### 4.2. Heterotopic Face Transplantation

We established microsurgical procedures for a heterotopic face transplantation model in mice [32] and describe them briefly as follows. For anesthesia, a mixture of ketamine and rocuronium was used in the donor mice and isoflurane in the recipient mice. The right side of the face from a donor was used to obtain the graft. Neck dissection was performed using a superfine microsurgical instrument and a microscope. The facial graft includes the upper and lower lips, right commissure, cheek (skin and mucosa), right hemi-mandible with two central incisors, masseter muscle, anterior facial vein, and carotid artery. The vascular pedicle of the facial graft contains the common carotid artery and anterior facial vein. The vessels are located next to each other at the inferior border of the mandible. The common carotid artery and posterior facial vein were utilized for recipient anastomosed vessels. Therefore, a total of 53 recipient C57BL/6 mice were transplanted for the evaluation of facial graft survival, as presented in Figure 1. In addition, another six recipient C57BL/6 mice were transplanted for the flow cytometry assay.

#### 4.2.1. Donor Surgery

The resection area on the right side of the face was marked with blue ink. The skin in the neck was incised in an oblique shape to expose the submandibular gland. This gland was retracted to expose the anterior facial vein and its branches. After ligating these branches meticulously, the gland was resected for wide exposure of the anterior facial vein. A hook was used to retract the sternocleidomastoid muscle laterally to expose all the other structures of the neck; then the common carotid artery was dissected. The posterior belly of the digastric muscle and the greater horn of the hyoid bone were removed, and the external carotid artery and its branches were exposed. Except for the facial artery, the internal carotid artery and all the branches of the external carotid artery were dissected and ligated. The vascular pedicle was released to the inferior border of the mandible. The facial anterior vein and common carotid artery were ligated proximally. Then a polyimide cuff was mounted by pulling the vessel through the cuff, everting the vessel over it, and securing it with a 10–0 nylon ligature. The skin of the flap was cut through the previous mark, including the upper lip, lower lip, and cheek, followed by a mandibular osteotomy including the incisor teeth; the floor of the mouth muscles and part of the cheek mucosa were also included. The dissection was carried out through the pterygoid muscles and blood vessels until visualization of zygomatic arch was achieved. The dissection was continued on the surface of the maxilla, dividing and exposing the infraorbital nerve. The whole right hemi-mandible with its condyle was disarticulated, including the masseter muscle. The parotid gland was removed, and the facial nerve was sheared. Then, the surrounding soft tissue to the zygomatic arch was separated, releasing the whole graft.

#### 4.2.2. Recipient Surgery

In the recipient mice, the neck skin was incised, the submandibular gland was released to expose the internal jugular vein to follow the anterior and posterior facial veins, and then the gland and the anterior facial vein were ligated. The facial posterior vein and external jugular vein were dissected to the inferior border of the mandible. A hook was used to retract the sternocleidomastoid muscle to the outside, to expose and dissect the common carotid artery until the inferior border of the posterior belly of the digastric muscle was ligated. The facial graft from the donor was fixed onto the cheek with nylon sutures, and anastomosis between the common carotid artery and the common carotid artery, and the anterior facial vein and the posterior facial vein was performed utilizing the cuff technique [60,61]. During the anastomosis using the cuff technique, heparinized saline was utilized for irrigation. The facial graft was checked to evaluate whether perfusion was adequate, and then the skin was sutured using 6–0 nylon sutures. The mean surgery time of the donor was 85 min (65–100 min), the mean ischemia time was approximately 90 min, and the mean surgery time of the recipient was 100 min (85–115 min).

### 4.3. Postoperative Care

The postoperative mice received antibiotic cephazolin (50 mg/kg/day), pain medicine (ketoprofen 1 mg/kg/day), and saline (30 mL/kg/day) by subcutaneous injection for 3 days. Allogeneic face transplanted mice received different dosages of tacrolimus (FK506, 0.6 and 2 mg/kg) via intraperitoneal injection for 21 days, beginning on postoperative day (POD) 1. FK506 is a macrocyclic polyketide that binds to the FK506 binding protein (FKBP) in T cells and then interacts with calcineurin to form the FK506–FKBP–calcineurin complex, which finally inhibits interleukin 2 production and T cell activation. Collected data of allograft survival showed the effects of the mandible on the prolongation of facial allograft survival. Weight measurement, imaging, and visual inspection were performed daily on each mouse. Rejection time referred to the time from the appearance of diffuse erythema or epidermolysis to complete necrosis of the transplanted allograft.

### 4.4. Histological Evaluation

Hematoxylin stains the nucleus by the staining with a blue-purple color. Eosin non-specially stains proteins with a pink color. Therefore, the infiltrating lymphocytes showed dense violet staining and the extracellular matrix indicated various degrees of pink staining in tissues [73,74]. Histological hematoxylin and eosin (H&E) staining showed lymphocyte infiltration into the facial allograft, which was attributed to the host’s alloimmunity against the graft. Facial flaps, including the lip skin and masseter muscle, were harvested on POD 14 and stained with H&E.

### 4.5. Flow Cytometry Analysis

Antibodies against cell surface markers were used to stain immune cells. For example, clusters of differentiation (CD4), CD8, CD19, natural killer cells (NK)1.1, CD11c, CD11b, CD25, H2D^d^, Forkhead box P3 (FoxP3), and interferon gamma (IFN-γ) were used for flow cytometry and were purchased from BD Biosciences (San Jose, CA, USA) and eBioscience (Waltham, MA, USA). For intracellular cytokine staining, phorbol 12-myristate 13-acetate (20 ng/mL), ionomycin (1 μg/mL), and monesine (4 μM) were used to stimulate immune cells for 4 h. Immune cells were analyzed using a BD FACSCanto™ II flow cytometer and CELLQUEST software (BD Biosciences, San Jose, CA, USA).

### 4.6. Statistical Analysis

All data are presented as mean ± standard deviation (SD). Statistical significance of facial survival rate was calculated utilizing the Kaplan–Meier method and compared for differences utilizing the log-rank test. The two-tailed Student’s *t* test was used to calculate significant differences between groups. All calculations were performed using GraphPad Prism 6 software. *p* < 0.05 was considered statistically significant.

## 5. Conclusions

The mandible has the potential to induce anti-inflammatory effects and mixed chimerism for prolonging facial allograft survival and may be correlated with the activity of MSCs and bone marrow hematopoietic stem cells. This study provides further insights into the effects of the mandible on the immunomodulatory processes in facial allograft survival. The mandible may facilitate reducing the use of immunosuppressive drugs when clinical facial allografts including the mandible are performed.

## Figures and Tables

**Figure 1 ijms-22-11104-f001:**
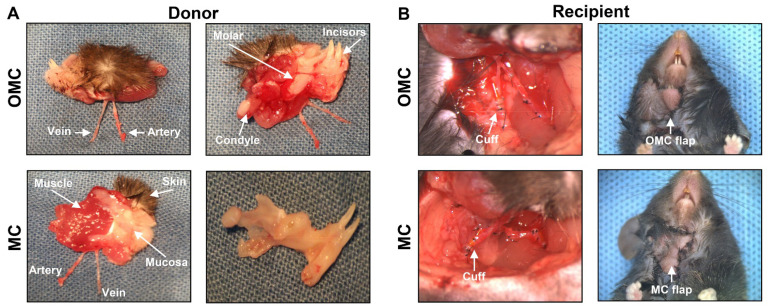
Surgery in the facial osteomyocutaneous (OMC) and myocutaneous (MC) groups. (**A**) Donor flap of the facial OMC group, including the mandible in addition to the skin, muscle, oral mucosa, and vessels. Donor flap of the facial MC group, containing only the skin, muscle, oral mucosa, and vessels, with the mandible removed. (**B**) The recipient’s vessels were anastomosed using the cuff technique for syngeneic facial OMC and MC transplantation.

**Figure 2 ijms-22-11104-f002:**
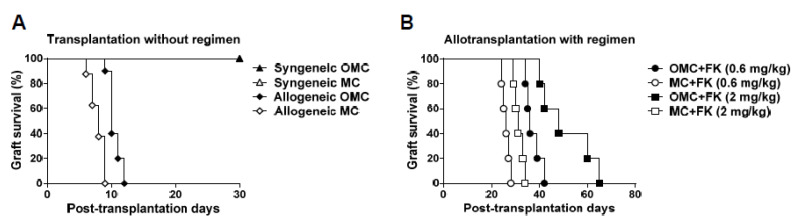
Survival of the facial flaps in the osteomyocutaneous (OMC) and myocutaneous (MC) allograft groups. (**A**) The survival rate of facial allografts in the OMC and MC graft groups without an immunosuppressive regimen. Fully major histocompatibility complex-incompatible BALB/c allografts were grafted onto the necks of C57BL/6 mice. The syngeneic group had facial grafts from C57BL/6 mice. The syngeneic OMC, syngeneic MC, allogeneic OMC, and allogeneic MC groups included a total of nine, six, ten, and eight mice, respectively. The Kaplan–Meier method was utilized to calculate the survival rate. The differences between the allogeneic OMC and MC groups were significant (*p* < 0.005). (**B**) Survival rate of facial flaps in the OMC and MC allograft groups with an immunosuppressive regimen. Two dosages of FK506 (0.6 and 2 mg/kg) were administered to transplanted mice via intraperitoneal injection for 21 days. The allogeneic OMC and MC groups receiving 0.6 mg/kg of FK506 had a total of five mice each. FK shown in (**B**) is the FK506 abbreviation. The allogeneic OMC and MC groups receiving 2 mg/kg of FK506 also had a total of five mice each. The differences between allogeneic OMC and MC groups for both dosages were significant (*p* < 0.005).

**Figure 3 ijms-22-11104-f003:**
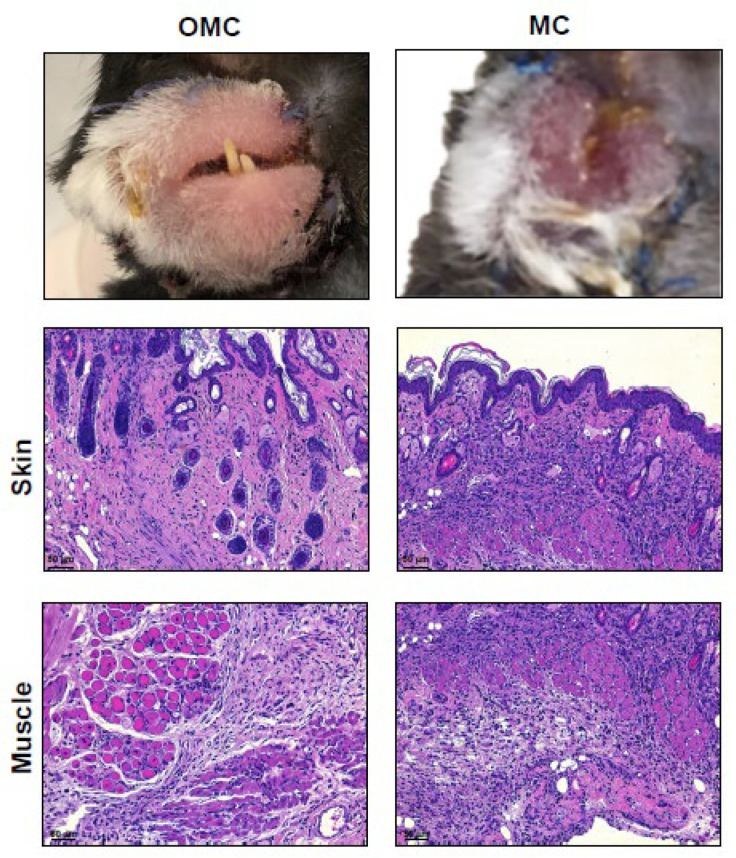
Degree of lymphocyte infiltration into facial allograft. Macroscopic and histological changes (indicated by hematoxylin and eosin staining) in facial allografts of osteomyocutaneous (OMC) and myocutaneous (MC) groups receiving 0.6 mg/kg of FK506 on postoperative day (POD) 14. Dense violet staining demonstrates infiltrating lymphocytes in the skin and muscle layers of the facial allograft. Scale bars are 50 μm.

**Figure 4 ijms-22-11104-f004:**
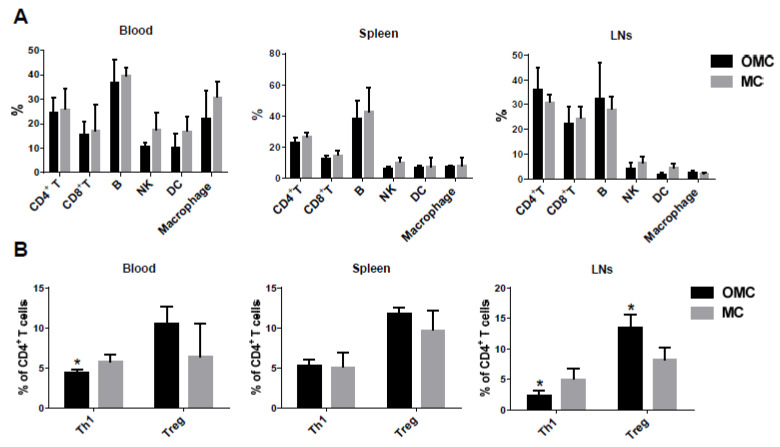
Development of leukocytes in face-allotransplanted mice. (**A**) Profiles of lymphocytes and monocytes in lymphoid organs of face-allotransplanted mice. Percentages of lymphocyte and monocyte subsets in the blood, spleen, and lymph nodes (LNs) of allogeneic osteomyocutaneous (OMC) and myocutaneous (MC) groups were examined using flow cytometry. Immune cells were harvested from lymphoid organs of mice from each group receiving 0.6 mg/kg of FK506 on POD 14. The immune cells were stained for cluster of differentiation (CD)4^+^ T and CD8^+^ T, CD19^+^ B, NK1.1^+^ nature killer cells (NKs), CD11c^+^ dendritic cells (DCs), CD11b^+^ macrophage cells. T, B, and NKs were gated from lymphocytes, and DCs and macrophages were gated from lymphocytes and monocytes for flow cytometry analysis. Data were collected from three mice in each group. The statistical data are presented as mean ± standard deviation (SD). (**B**) Profiles of pro-inflammatory T helper (Th)1 and anti-inflammatory regulatory T (Treg) cells in lymphoid organs of face-allotransplanted mice. Percentages of pro-inflammatory Th1 and anti-inflammatory Treg cell subsets in the blood, spleen, and lymph nodes of allogeneic OMC and MC groups were analyzed using flow cytometry. Lymphocytes were isolated from lymphoid organs of mice from each group receiving 0.6 mg/kg of FK506 on POD 14 for CD4^+^ T cell subset analysis. The immune cells were stained for CD4^+^IFN-γ^+^ (interferon gamma) Th1 and CD4^+^CD25^+^ FoxP3^+^ (Forkhead box P3) Treg cells. Data were collected from three mice in each group. The statistical data are presented as mean ± SD. The allogeneic OMC group was compared with the MC group for each lymphoid organ (* *p* < 0.05, Student’s *t* test).

**Figure 5 ijms-22-11104-f005:**
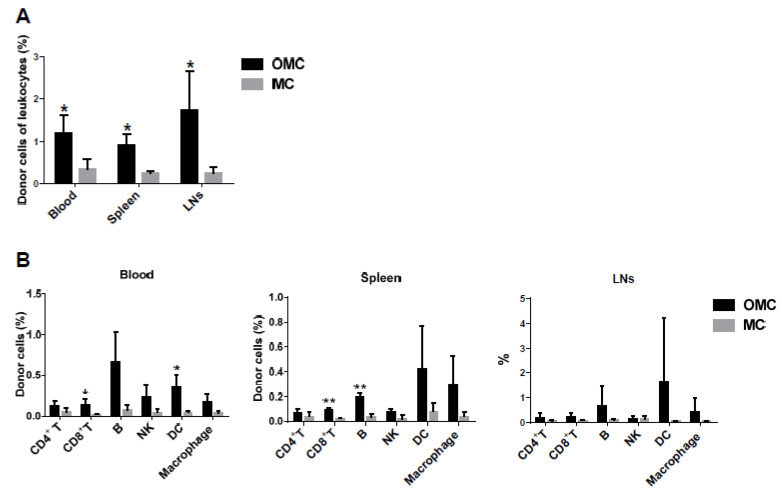
Mixed chimerism in lymphoid organs of face-allotransplanted mice. (**A**) Percentages of donor cells (H2^d^) in the blood, spleen, and lymph nodes (LNs) of allogeneic osteomyocutaneous (OMC) and myocutaneous (MC) groups were analyzed using flow cytometry. Immune cells were isolated from the lymphoid organs of mice from each group receiving 0.6 mg/kg of FK506 on POD 14. The donor cells were stained for H2^d+^ cells. Data were collected from three mice in each group. The statistical data are presented as mean ± SD. (**B**) Multilineage chimerism developments in lymphoid organs of face-allotransplanted mice. Percentages of donor cells (H2^d^) in the blood, spleen, and lymph nodes of allogeneic OMC and MC groups were examined by flow cytometry. Immune cells were isolated from the lymphoid organs of mice from each group receiving 0.6 mg/kg of FK506 on POD 14. The multilineage donor cells were stained for H2^d+^CD4^+^ T and H2^d+^CD8^+^ T, H2^d+^CD19^+^ B, H2^d+^NK1.1^+^ NKs, H2^d+^CD11c^+^ DCs, H2^d+^CD11b^+^ macrophage cells. T, B, and NKs were gated from lymphocytes, and DCs and macrophages were gated from lymphocytes and monocytes for flow cytometry analysis. Data were collected from three mice in each group. The statistical data are expressed as mean ± SD. The allogeneic OMC group was compared with the MC group for each lymphoid organ (* *p* < 0.05 and ** *p* < 0.05, Student’s *t* test).

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
