# Peer review of "The Mandible Ameliorates Facial Allograft Rejection and Is Associated with the Development of Regulatory T Cells and Mixed Chimerism"

_ijms, 2021, doi:10.3390/ijms222011104_

Round 1

Reviewer 1 Report

Dear Author,

I invite you to increase the introduction by adding more recent articles regarding the self bone healing in maxillo-facial traumatology, dental implantology, orofacial anomalies and in the most frequent craniofacial cases.

these articles could be useful:

  • Unilateral superior condylar neck fracture with dislocation in a child treated with an acrylic splint in the upper arch for functional repositioning of the mandible PMID: 27398739
  • Can bone compaction improve primary implant stability? An in vitro comparative study with osseodensification technique DOI:10.3390/app11083427
  • Polyphenols as potential agents in the management of temporomandibular disordersDOI: 10.3390/APP10155305
  • Mandibular coronoid process hypertrophy: Diagnosis and 20-year follow-up with CBCT, MRI and EMG evaluations DOI: 10.3390/app11104504
  • Oral-facial-digital syndrome (OFD): 31-year follow-up management and monitoring PMID: 29460530
    Best regards

Author Response

REVIEWER 1

Dear Author,

I invite you to increase the introduction by adding more recent articles regarding the self bone healing in maxillo-facial traumatology, dental implantology, orofacial anomalies and in the most frequent craniofacial cases.

these articles could be useful:

  • Unilateral superior condylar neck fracture with dislocation in a child treated with an acrylic splint in the upper arch for functional repositioning of the mandible PMID: 27398739
  • Can bone compaction improve primary implant stability? An in vitro comparative study with osseodensification technique DOI:10.3390/app11083427
  • Polyphenols as potential agents in the management of temporomandibular disordersDOI: 10.3390/APP10155305
  • Mandibular coronoid process hypertrophy: Diagnosis and 20-year follow-up with CBCT, MRI and EMG evaluations DOI: 10.3390/app11104504
  • Oral-facial-digital syndrome (OFD): 31-year follow-up management and monitoring PMID: 29460530

Ans: Thank you very much for your comments. We have added this reference information into introduction part in line 44~59.

In addition, we also revised the manuscript by the MDPI English editing.

REVIEWER 2

The paper is unappropriately written, here are some suggestions of correction, although the whole paper requires rewriting - therefore I suggest to reject it

  1. The abstract refers to the text

Ans: Thank you very much for your comments. We have added more result information in the abstract, which was highlighted by yellow color.

  1. Introduction could be more detailed - there should be some information concerning the division of types of grafts (maybe in a table?)

Ans: Thank you very much for your comments. Apology for this.

(1) We have added more information in the part of introduction (line 44~59; 68~72; 74~76; 77~83; 87~91).

(2) We have added more information about various types of composite grafts in line 77~82.

  1. The Material and Methods are presented in 4th chapter, which is unacceptable - this should be 2nd section and should be more detailed, eg. there is no information how many mice were there used for the study. There is no information when and where were the studies performed (Authors from 3 countries, 2 continents)

Ans: Thank you very much for your comments.

(1) We have moved the part of materials and methods to 2nd section.

(2) In fact, we described mice numbers used in different experiments in the figure legends (line 241, 246, 247, 283, 288, 320, 326). In order to make the information clearer, we have added total mice numbers for surgery and flow cytometry in the Method part according to the Reviewer’s comments (Line 126~129).

(3) The whole study was performed in Laboratory Animal Center for animal care or Research Building, Chang Gung Memorial Hospital, Taoyuan, Taiwan. The first three authors including Dr. De Paz, Dr. Avina, and Dr. Cardona, all are clinical and research fellows from Peru, Mexico, and Colombia countries, respectively. These fellows came to our Department of Plastic Surgery, Chang Gung Memorial Hospital, Taiwan for training animal surgery, research study, and clinical observation. they all spent two years in our center for vascularized composite allotransplantation for training and study. Dr. Cardon established the mouse facial transplantation model and taught Dr. De Paz how to operate the mouse facial surgery because of only a short time overlap between them. Dr. De Paz performed all surgery and most experimental assays. Dr. Avina helped Dr. De Paz perform the postoperative care including pain medicine injection and observe the rejection time of transplanted mice because of one year overlap between them. In addition to the first three authors, other authors all come from Taiwan.

(4) We also added other information in the part of Method in line 178~181 and 191~192.

  1. The figures in results section are unclear, there is no explanation to them neither in the main text, nor at the figure title - at least the most important numbers should be in the text body.

Ans: Thank you very much for your comments. We have added more information in results and figure legends in line 206~208; 212~213. We also revised the results and legends, which were highlighted by yellow color. Figure numbers were also mentioned in the part of results.

  1. Discussion should be more detailed

Ans: Thank you very much for your comments. We have added more information in line 341~352.

  1. More than half of the references is from the past 10 years, although at least half of the references should refer to the past 5 years (most of them should refer to the past 10 years).

Ans: Thank you very much for your comments. We have increased more than half of the references from the past 5 years (37 references from the past 5 years/70 total references=52.8%).

In addition, we also revised the manuscript by the MDPI English editing.

Reviewer 2 Report

The paper is unappropriately written, here are some suggestions of correction, although the whole paper requires rewriting - therefore I suggest to reject it

  1. The abstract refers to the text
  2. Introduction could be more detailed - there should be some information concerning the division  of types of grafts (maybe in a table?)
  3. The Material and Methods are presented in 4th chapter, which is unacceptable - this should be 2nd section and should be more detailed, eg. there is no information how many mice were there used for the study. There is no information when and where were the studies performed (Authors from 3 countries, 2 continents)
  4. The figures in results section are unclear, there is no explenation to them neither in the main text, nor at the figure title - at least the most important numbers should be in the text body.
  5. Discussion should be more detailed
  6. More than half of the references is from the past 10 years, although at least half of the references should refer to the past 5 years (most of them should refer to the past 10 years).

Best regards

Author Response

(The authors gave the same response as above.)

Round 2

Reviewer 2 Report

Dear Sirs,

thank you for the new version, I still have some suggestions to be changed:

  1. When writing of physiotherapy in TMJ, please quote Wieckiewicz M as an expert in this field, eg. - Boening K, Wieckiewicz M, Paradowska-Stolarz A, Wiland P, Shiau YY. Temporomandibular disorders and oral parafunctions: mechanism, diagnostics, and therapy. Biomed Res Int. 2015;2015:354759. doi: 10.1155/2015/354759 or Miernik M, Wieckiewicz M, Paradowska A, Wieckiewicz W. Massage therapy in myofascial TMD pain management. Adv Clin Exp Med. 2012 Sep-Oct;21(5):681-5.
  2. When writing of facet transplants it is good that you quoted prof. Siemionov, but it would be valid to quote Rodriguez as well - Rifkin WJ, Manjunath AK, Kantar RS, Jacoby A, Kimberly LL, Gelb BE, Diaz-Siso JR, Rodriguez ED. A Comparison of Immunosuppression Regimens in Hand, Face, and Kidney Transplantation. J Surg Res. 2021 Feb;258:17-22. doi: 10.1016/j.jss.2020.08.006. or Mills EC, Alfonso AR, Wolfe EM, Park JJ, Sweeney GN, Hoffman AF, Felsenheld JH, Sosin M, Ramly EP, Rodriguez ED. Public Perceptions of Cross-Sex Vascularized Composite Allotransplantation. Ann Plast Surg. 2020 Dec;85(6):685-690. doi: 10.1097/SAP.0000000000002472. or Diep GK, Ramly EP, Alfonso AR, Berman ZP, Rodriguez ED. Enhancing Face Transplant Outcomes: Fundamental Principles of Facial Allograft Revision. Plast Reconstr Surg Glob Open. 2020 Aug 17;8(8):e2949. doi: 10.1097/GOX.0000000000002949. PMID: 32983759; 
  3. I would also add some information regarding why allografts are the best solution when bone is missing in the introduction
  4. Please check if all the abreviations are translated in the text. I would also add explanations of them then they occur at the figure (in the description - title of the figure)
  5. Please, add limitations of the study (eg. small number, mice) and perspectives for the future in the discussion session.

After that, paper could be reconsidered to be published, as the Authors did a huge amout of work in reconstructing the paper before.

Author Response

REVIEWER 2

thank you for the new version, I still have some suggestions to be changed:

  1. When writing of physiotherapy in TMJ, please quote Wieckiewicz M as an expert in this field, eg. - Boening K, Wieckiewicz M, Paradowska-Stolarz A, Wiland P, Shiau YY. Temporomandibular disorders and oral parafunctions: mechanism, diagnostics, and therapy. Biomed Res Int. 2015;2015:354759. doi: 10.1155/2015/354759 or Miernik M, Wieckiewicz M, Paradowska A, Wieckiewicz W. Massage therapy in myofascial TMD pain management. Adv Clin Exp Med. 2012 Sep-Oct;21(5):681-5.

Ans: Thank you very much for your comments. We have added the two references in the introduction (line 54~55).

  1. When writing of facet transplants it is good that you quoted prof. Siemionov, but it would be valid to quote Rodriguez as well - Rifkin WJ, Manjunath AK, Kantar RS, Jacoby A, Kimberly LL, Gelb BE, Diaz-Siso JR, Rodriguez ED. A Comparison of Immunosuppression Regimens in Hand, Face, and Kidney Transplantation. J Surg Res. 2021 Feb;258:17-22. doi: 10.1016/j.jss.2020.08.006. or Mills EC, Alfonso AR, Wolfe EM, Park JJ, Sweeney GN, Hoffman AF, Felsenheld JH, Sosin M, Ramly EP, Rodriguez ED. Public Perceptions of Cross-Sex Vascularized Composite Allotransplantation. Ann Plast Surg. 2020 Dec;85(6):685-690. doi: 10.1097/SAP.0000000000002472. or Diep GK, Ramly EP, Alfonso AR, Berman ZP, Rodriguez ED. Enhancing Face Transplant Outcomes: Fundamental Principles of Facial Allograft Revision. Plast Reconstr Surg Glob Open. 2020 Aug 17;8(8):e2949. doi: 10.1097/GOX.0000000000002949. PMID: 32983759; 

Ans: Thank you very much for your comments. We have added the three references (R24, 25, 26) in the introduction (line 64 & 80).

  1. I would also add some information regarding why allografts are the best solution when bone is missing in the introduction

Ans: Thank you very much for your comments. We have added some information in introduction according to reviewer’s comments (line 61~62).

  1. Please check if all the abbreviations are translated in the text. I would also add explanations of them then they occur at the figure (in the description - title of the figure)

Ans: Thank you very much for your comments. We have checked and added abbreviation descriptions in the manuscript and figure legends (line 176~178; 193~194; 207; 245, 253; 266~267; 288, 290, 293, 294, 298; 328)

  1. Please, add limitations of the study (eg. small number, mice) and perspectives for the future in the discussion session.

Ans: Thank you very much for your comments. We have added a paragraph about the comment in discussion (line 402~417).

After that, paper could be reconsidered to be published, as the Authors did a huge amout of work in reconstructing the paper before.

Round 3

Reviewer 2 Report

Dear Sirs, paper could be accepted in this form